# InfiR2: A Comprehensive FP8 Training Recipe for Reasoning-Enhanced Language Models

## Abstract

The immense computational cost of training Large Language Models (LLMs) presents a major barrier to innovation. While FP8 training offers a promising solution with significant theoretical efficiency gains, its widespread adoption has been hindered by the lack of a comprehensive, open-source training recipe. To bridge this gap, we introduce an end-to-end FP8 training recipe that seamlessly integrates continual pre-training and supervised fine-tuning. Our methodology employs a fine-grained, hybrid-granularity quantization strategy to maintain numerical fidelity while maximizing computational efficiency. Through extensive experiments, including the continue pre-training of models on a 160B-token corpus, we demonstrate that our recipe is not only remarkably stable but also essentially lossless, achieving performance on par with the BF16 baseline across a suite of reasoning benchmarks. Crucially, this is achieved with substantial efficiency improvements, including up to a 22% reduction in training time, a 14% decrease in peak memory usage, and a 19% increase in throughput. Our results establish FP8 as a practical and robust alternative to BF16, and we will release the accompanying code to further democratize large-scale model training.

## 1 Introduction

The emergence of Large Language Models (LLMs) is revolutionizing artificial intelligence, demonstrating unprecedented capabilities in tasks like natural language processing, code generation, and multimodal reasoning (OpenAI et al., 2024; Team et al., 2024; DeepSeek-AI et al., 2025; Qwen et al., 2025). However, this progress is fundamentally tethered to scaling laws that demand immense computational resources, rendering the training of these models prohibitively expensive and creating a significant barrier to innovation. To surmount this challenge, researchers are actively pursuing more efficient training paradigms, with low-precision training emerging as a highly promising direction (Micikevicius et al., 2018; Wang et al., 2018a; Zhu et al., 2019; Xi et al., 2023; Wortsman et al., 2023; Xi et al., 2024).

Low-precision training accelerates computation and reduces memory usage by quantizing network tensors to lower bit formats. While BF16 is the current de facto standard widely adopted in large-scale model training (Kalamkar et al., 2019), a paradigm shift is underway. The advent of NVIDIA's Hopper architecture, with its dedicated hardware support for FP8, has unlocked new opportunities for efficiency. FP8 training offers the compelling theoretical potential to double training throughput and halve the memory footprint relative to BF16.

Pioneering work by DeepSeek-AI et al. (2025) provided a crucial existence proof for the viability of FP8 training. They introduced a fine-grained quantization strategy for efficient General Matrix Multiplication (GEMM), open-sourcing their implementation as DeepGEMM (deepseek ai, 2024), and demonstrated its application in reducing memory and communication overhead for Mixture-of-Experts (MoE) models. While this work successfully showcased the potential of FP8, the absence of an open-source, end-to-end training recipe has remained a critical gap, hindering widespread community adoption and further research into the nuances of FP8 training dynamics.

To bridge this gap and democratize FP8 training, we introduce a comprehensive FP8 training recipe that seamlessly integrates FP8 techniques with mainstream LLM training paradigms, including pre-training and supervised fine-tuning. Our core methodology is a hybrid-granularity quantization

strategy: we apply computationally-friendly block-wise quantization to model weights, while employing higher-fidelity token-wise quantization for activations, which are known to exhibit more dynamic ranges. Based on this end-to-end FP8 training recipe, we obtain InfiR2-1.5B-FP8 and InfiR2-7B-FP8.

Through extensive experiments in both continual pre-training and supervised fine-tuning settings, we demonstrate the efficacy and efficiency of our recipe. Our results show that end-to-end FP8 training is remarkably stable, with training and validation loss curves that are nearly identical to those of BF16. Furthermore, we find that FP8 training is largely lossless, achieving performance on par with the BF16 baseline across a suite of reasoning benchmarks, and in some cases, even yielding performance improvements, particularly in smaller-scale models. Crucially, these results are achieved alongside substantial gains in training efficiency. Our FP8 implementation reduces total training time by up to 22%, decreases peak memory consumption by up to 14%, and improves computational throughput by up to 19%. Collectively, our work validates that FP8 training, facilitated by our recipe, offers a compelling trade-off, delivering significant improvements in training speed and resource utilization without compromising model performance, thereby making the development of large-scale models more accessible and sustainable.

Our contributions are as follows:

- We introduce an FP8 fine-grained hybrid granularity training recipe for stable and efficient large-scale model training, specifically designed to preserve and enhance model reasoning capabilities.

- We provide rigorous empirical validation of FP8 training accuracy across continual pre-training and SFT, showing that FP8 matches BF16 with performance differences typically within 1–2 points across benchmarks, while delivering up to 22% faster training, 14% lower memory footprint, and 19% higher throughput.

- We release valuable insights, training logs, and intermediate model checkpoints to support the community, including the InfiR2-7B-FP8 model that achieves 55.73 on AIME24 (+12.71 over its Qwen2.5-7B-base origin).

## 2 RELATED WORK

**Low-precision training in LLMs** Recent advancements in low-precision training have shown promise in mitigating the substantial computational and memory demands of modern deep neural networks (Hubara et al., 2016; Wang et al., 2018b; Zhou et al., 2016; Wu et al., 2018). Despite this, applying low precision to the training of LLMs remains a significant challenge. A key obstacle lies in the strong outliers that emerge in the activations of modern transformer models (Lin et al., 2024; smo; Kovaleva et al., 2021; Bondarenko et al., 2021; Dettmers et al., 2022). This phenomenon, which complicates quantization, is primarily attributed to the combination of the softmax, residual blocks and LayerNorm, posing a fundamental barrier to the effective use of low-precision techniques for LLM training (Bondarenko et al., 2023).

To address these issues and enable low-precision training, the community has largely relied on mixed-precision techniques. FP16, addressing the information loss of the half-precision by introducing the loss scaling, becomes one of the most prevalent low-precision training methods for LLMs (Micikevicius et al., 2017). An evolution of this approach, BF16 training (Hubara et al., 2016) provides a wider representation range than FP16. This wider exponent range lets it handle the large gradients and activation outliers common, and therefore more stable in LLMs training. However, both methods compute the forward and backward passes in half-precision, while maintaining master weights, gradients, and optimizer states in FP32 for numerical stability.

As models continue to grow in scale, the field has also begun exploring even lower-precision training methods, with FP8 training drawing significant attention (Fishman et al., 2025; Wang et al., 2018b; Peng et al., 2023). On the hardware side, NVIDIA's Blackwell architecture (NVIDIA, 2024) fundamentally ensures FP8 training stability with the MXFP8-E4M3 format (Rouhani et al., 2023) and a quantization method that preserves powers of two for scaling factors. Together, these innovations are pushing the boundaries of low-precision training, making it an indispensable part of large model research.

However, this lower precision introduces new challenges, such as a greater risk of precision loss and training instability (Lee et al., 2024). To overcome these hurdles, COAT framework (Xi et al., 2025) uses techniques like dynamic range expansion and mixed-granularity activation quantization to preserve accuracy. DeepSeek-V3 (DeepSeek-AI et al., 2025), meanwhile, employs a fine-grained quantization strategy that meticulously groups and scales weights and activations to handle outliers.

# 3 PRELIMINARIES

**FP8** has become a key technique for improving the efficiency of large models during both training and Inference. Its main advantages include reducing memory footprint and accelerating computation. Micikevicius et al. (2022) standardized two FP8 formats for deep learning: E4M3 (4 exponent bits, 3 mantissa bits), which has a smaller dynamic range but higher precision and is suitable for weights, and E5M2 (5 exponent bits, 2 mantissa bits), which has a wider dynamic range and is better suited for activations.

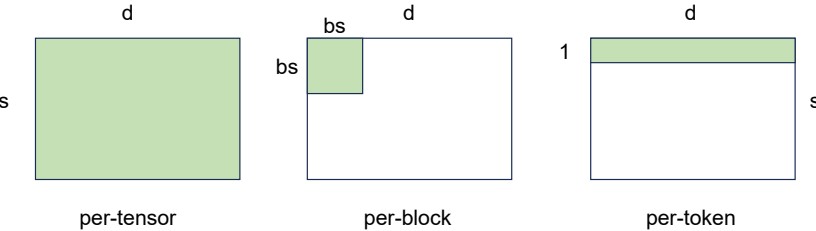

Figure 1: An illustration of three common quantization granularities: per-tensor, per-block, and per-token. The tensor has a shape of [s, d], where s is the context length and d is the dimension. bs represents the block size.

**Quantization** is the process of mapping a high-precision tensor $X$ to low-precision representation, typically using a scaling factor ($S$). Based on the granularity at which the scaling factor is applied, there are three primary quantization methods, as illustrated in Figure 1:

- **Per-Tensor:** A single scaling factor is applied across the entire tensor. While computationally simple, this method is prone to accuracy loss from outlier values.
- **Per-Block:** The tensor is partitioned into smaller blocks (e.g., bs×bs sub-matrices), each with a unique scaling factor. This method offers a better balance between computational efficiency and numerical precision.
- **Per-Token:** This fine-grained method applies a distinct scaling factor to each individual token (or row). Specifically, this can also be represented as a 1 × bs block, where bs is the block size. While this approach maximizes accuracy, it comes at a higher computational cost.

All three quantization methods follow a simple two-step process. First, a scaling factor (S) is calculated. This is done by taking the largest absolute value in a given tensor (or its sub-part) and dividing it by the maximum value that the FP8 format can represent (Vmax).

$$S = \frac{\max(|X|)}{V_{\max}} \tag{1}$$

Next, this scaling factor is used to convert each number (x) in the original tensor (X) into its new quantized value (Q (x)) by rounding the result.

$$Q(x) = \text{round}\left(\frac{x}{S}\right) \tag{2}$$

**Mixed-Granularity Quantization**: To address the inefficiency and inaccurate problem, COAT (Xi et al., 2025) propose to use mixed granularity FP8 precision flow to improve the accuracy without

introducing too much overhead. For linear layers, they apply per-tensor quantization to maximize the performance of Tensor Cores. instead of using per-block quantization with block size bs × bs as proposed in Fishman et al. (2025), they propose to use per-token quantization with size 1 × G, where $G = bs^2$ to keep the granularity the same.

**UE8M0.** proposed by Mishra et al. (2025), ensures the stability of post-quantization FP8 training by rounding the scaling factor X up to the nearest power of 2. As detailed in Algorithm 1, this upward rounding guarantees the scaling factor is slightly greater than or equal to its theoretical value, which in turn maps a wide numerical range into the limited 8-bit representation without introducing significant quantization noise.

---

**Algorithm 1** Scaling factor $X$ computation method for UE8M0 quantization format.

---

1: $X_{\text{float}} \leftarrow a_{\max}/d_{\max}$       ▷ $d_{\max}$ is the max representable value in the MX-format
2: $\text{expX}_{\text{float}} \leftarrow \log_2(X_{\text{float}})$       ▷ Extract the de-biased exponent
3: $\text{expX}_{\text{int}} \leftarrow \lceil \text{expX}_{\text{float}} \rceil$       ▷ **Round up**
4: $X \leftarrow \text{clamp}(\text{expX}_{\text{int}}, -127, 127)$       ▷ clamp to the E8M0 representable exponent range
5: $X \leftarrow X + 127$       ▷ Add bias
6: **return** $2^X$

---

## 4   HYBRID GRANULARITY QUANTIZATION STRATEGY

To address the dual challenges of Out-of-Memory (OOM) errors and accuracy degradation in large-scale model training, we adopt a hybrid granularity quantization strategy in FP8 training, applying different quantization methods to weights and activations, as shown in Figure 2.

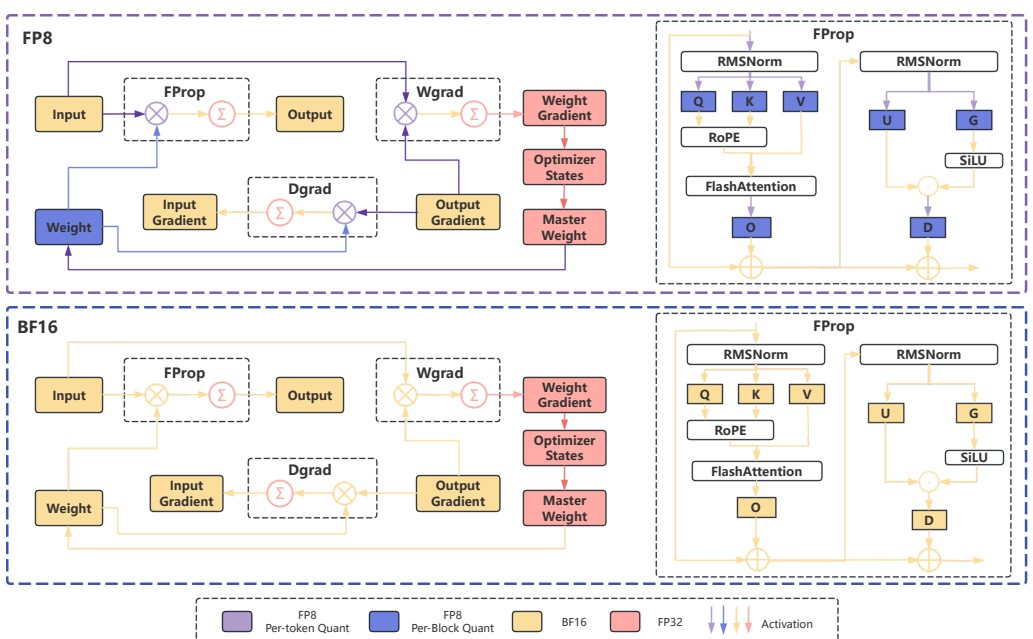

Figure 2: An illustration of a hybrid granularity quantization strategy using FP8, compared to a standard BF16 pipeline. In the FP8 pipeline, different quantization methods are applied: per-tensor quantization for weights (purple), and per-block quantization for activations (blue). The diagram shows the complete training process, including forward propagation (FProp), weight gradient calculation (Wgrad), and input gradient calculation (Dgrad), along with a detailed view of the FProp workflow.

**Weight Quantization**: To align with hardware-level optimizations for Deep GEMM operations, we implement block-wise quantization for model weights. Compared to COAT which uses the coarser

tensor-wise method, this approach maintains a higher degree of precision while still providing a substantial reduction in memory consumption, effectively alleviating resource bottlenecks during training.

**Activation Quantization**: For activations, particularly those preceding the SwiGLU activation function, we employ a more granular token-wise quantization. This method is better suited to the unique numerical range of each token, maximizing the preservation of critical information.

To guarantee the stability and accuracy of the training process, we maintain three critical components in high-precision FP32. The weight gradient and optimizer states are kept in FP32 to ensure the preservation of small, precise values that would otherwise be lost to rounding errors in a lower-precision format. Similarly, the master weight is stored in FP32 as a high-fidelity reference copy, accumulating small updates from the gradients without information loss, thereby preventing catastrophic degradation in model quality over time.

## 5 EXPERIMENTS

In this section, we present our experiments on pre-training and supervised fine-tuning (SFT) with FP8, covering the experimental setup, training results, efficiency analysis, and key findings.

### 5.1 PRETRAINING AND FINE-TUNING WITH FP8

#### 5.1.1 SETUP

We perform continual pre-training on the Qwen2.5-1.5B-base and Qwen2.5-7B-base models for an additional 160B tokens using an FP8 format, where both forward and backward passes employ E4M3, and quantization scaling factors are represented in UE8M0. The data mixture consists of 140B tokens from public sources (FineWeb (Penedo et al., 2024), Nemotron Datasets (Chowdhery et al., 2024), stack-edu (Kocetkov et al., 2023) and issues-kaggle-notebooks (Lozhkov et al., 2024)) and a subsequent 20B tokens where this base is mixed with AM-DeepSeek-R1 (a-m team, 2025) and AM-Qwen3 (Tian et al., 2025). All pre-training hyperparameters are listed in Table 1.

Subsequently, these models are fine-tuned with FP8 format in two stages using the InfiAlign-SFT-92k (Cai et al., 2025) and InfiAlign-SFT-165k datasets, with hyperparameters shown in Table 3. This stage yields InfiR2-1.5B-FP8 and InfiR2-7B-FP8. Finally, we evaluate them on the AIME24, AIME25 (Trinh et al., 2024), GPQA (Rein et al., 2023), and LiveCodeBench v5 (Cai et al., 2024) benchmarks using the EvalScope framework (ModelScope). To ensure a fair and consistent assessment, the same precision is maintained throughout both the training and evaluation phases.

Table 1: Continual Pretrain Configuration

| Hyperparameter | Value |
|---|---|
| Batch Size | 128 |
| Learning Rate | $1 \times 10^{-4}$ |
| Minimun Learning Rate | $1 \times 10^{-5}$ |
| Weight decay | 0.1 |
| Context Length | 32k |

#### 5.1.2 TRAINING LOSS

As shown in 4, the trajectories of the FP8 (red) and BF16 (blue) curves are nearly identical throughout the entire 160B token training process, for both the validation loss (a) and the training loss (b). The two curves are so closely aligned they are virtually indistinguishable. This indicates that the model's learning process is equivalent under both precision formats.

#### 5.1.3 TRAINING RESULTS

Our end-to-end FP8 training recipe demonstrates strong effectiveness in enhancing foundation models. As shown in Table 2, both 1.5B and 7B models trained with InfiR2 achieve substantial gains

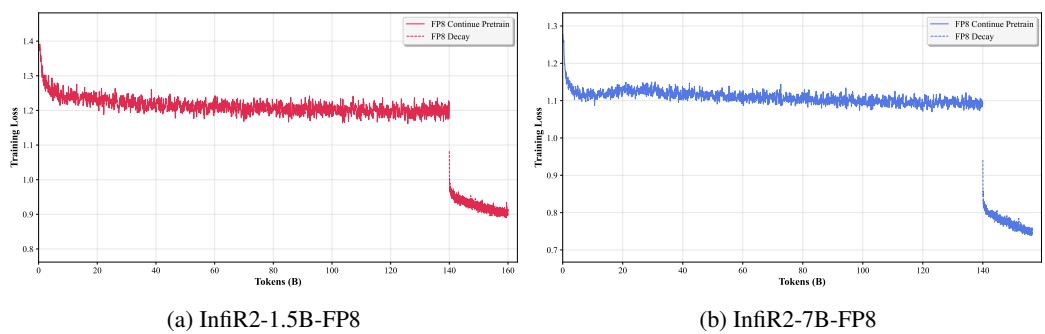

(a) InfiR2-1.5B-FP8        (b) InfiR2-7B-FP8

Figure 3: The FP8 training loss of InfiR2-1.5B and InfiR2-7B.

over their Qwen2.5-base counterparts. At the 1.5B scale, InfiR2-1.5B-FP8 consistently surpasses the base model with the same supervised fine-tuning across all benchmarks. The improvements become even more pronounced at the 7B scale: InfiR2-7B-FP8 attains a score of 55.73 on AIME 24, representing a 12.71-point increase over its base and significantly outperforming the strong DeepSeek-Distill-Qwen-7B baseline. These results confirm that our FP8 training pipeline produces high-quality, competitive models with superior reasoning capabilities.

Table 2: Performance comparison on reasoning benchmarks. All scores are percentages (%).

| Model | AIME 25 | AIME 24 | GPQA | LiveCodeBench v5 |
|---|---|---|---|---|
| Deepseek-Distill-Qwen-1.5B | 21.35 | 26.87 | 32.26 | 18.50 |
| Qwen2.5-1.5B-base (w. InfiAlign) | 14.58 | 10.52 | 28.98 | 12.99 |
| InfiR2-1.5B-FP8 | 18.45 | 17.39 | 29.48 | 17.10 |
| Deepseek-Distill-Qwen-7B | 43.00 | 49.00 | 48.20 | 37.60 |
| Qwen2.5-7B-base (w. InfiAlign) | 33.75 | 43.02 | 48.11 | 39.48 |
| InfiR2-7B-FP8 | 40.62 | 55.73 | 45.33 | 40.31 |

## 5.2 SUPERVISED FINE-TUNING WITH FP8

### 5.2.1 SETUP

We perform a two-stage Supervised Fine-Tuning on the Qwen2.5-Math-7B and Qwen2.5-Math-1.5B models to compare the performance across BF16, FP8(w. FP32 scale), and FP8. The key training hyperparameters are summarized in Table 3.

Table 3: SFT Configuration

| Hyperparameter | Value |
|---|---|
| Stage1 Dataset | InfiAlign-SFT-92k |
| Stage2 Dataset | InfiAlign-SFT-165k |
| Epochs | 5 |
| Batch Size | 64 |
| Learning Rate | $5 \times 10^{-5}$ |
| Context Length | 32k |
| LR | Cosine with 0.1 warm-up ratio |

### 5.2.2 TRAINING RESULTS

The performance of Qwen2.5-Math-1.5B and Qwen2.5-Math-7B with different quantization methods is shown in Table 4.

Table 4: A detailed comparison of training quantization methods (BF16, FP8 w. FP32 scale, FP8) on the performance of Qwen2.5-Math-7B and Qwen2.5-Math-1.5B. The models were evaluated on AIME 25, AIME24, GPQA, and LiveCodeBench v5 using checkpoints from Stage 1 and Stage 2 of training.

| Base Model | Quantization Method | AIME 25 | AIME24 | GPQA | LiveCodeBench v5 |
|---|---|---|---|---|---|
| | *Stage 1* | | | | |
| | BF16 | 44.16 | 56.87 | 45.14 | 32.22 |
| | FP8 w. FP32 scale | 44.06 | 56.67 | 47.98 | 32.18 |
| | FP8 | 44.89 | 57.81 | 47.10 | 31.34 |
| **Qwen2.5-Math-7B** | *Stage 2* | | | | |
| | BF16 | 50.00 | 59.48 | 48.36 | 35.22 |
| | FP8 w. FP32 scale | 46.46 | 57.92 | 45.39 | 35.87 |
| | FP8 | 49.79 | 59.69 | 46.78 | 36.21 |
| | *Stage 1* | | | | |
| | BF16 | 15.41 | 18.33 | 24.68 | 10.71 |
| | FP8 w. FP32 scale | 15.73 | 18.65 | 25.38 | 10.14 |
| | FP8 | 17.50 | 16.88 | 23.17 | 9.84 |
| **Qwen2.5-Math-1.5B** | *Stage 2* | | | | |
| | BF16 | 17.92 | 21.35 | 24.48 | 12.16 |
| | FP8 w. FP32 scale | 20.62 | 22.81 | 27.78 | 12.69 |
| | FP8 | 20.73 | 21.77 | 25.13 | 12.96 |

### 5.2.3 FINDINGS

**FP8 Quantization: Performance Preservation with Occasional Gains.** We find that FP8 quantization largely maintains performance fidelity relative to the BF16 baseline. Across most benchmarks, the performance metrics for FP8 are on par with BF16, with observed deltas typically confined to a 1-2 point margin, a variance attributable to inherent evaluation noise.

**UE8M0 better than FP32 scale.** For the Qwen2.5-Math-7B model, all precision formats demonstrate comparable performance in Stage 1. For instance, on the AIME24 benchmark, FP8 (57.81) registers a slight improvement over BF16 (56.87). Although FP8(w. FP32 scale) exhibits a marginal performance regression in Stage 2 on select benchmarks (e.g., 46.46 vs. 50.00 for BF16 on AIME 25), the FP8 configuration effectively mitigates this effect, restoring performance to a level (49.79) nearly identical to the BF16 baseline. This result underscores the criticality of the UE8M0 format for preserving model performance in larger-scale models during advanced training stages.

**Performance Gains in Smaller-Scale Models.** Counter-intuitively, for the smaller-scale Qwen2.5-Math-1.5B model, both FP8 quantization methods yield substantial performance improvements over the BF16 baseline in Stage 2. On the GPQA benchmark, for example, the FP8(w. FP32 scale) and FP8 configurations achieve scores of 27.78 and 25.13, respectively, compared to 24.48 for BF16. Notably, the FP8(w. FP32 scale) variant consistently outperforms the baseline across all four evaluated benchmarks.

## 5.3 FP8 TRAINING VALIDATION

To validate the accuracy alignment of our FP8 training recipe with a BF16 baseline, we conduct a two-stage comparison on the Qwen2.5-1.5B-base model. First, we compare the loss curves during the continual pre-training phase for both precisions with settings described in Section 5.1. Second, we perform a comprehensive evaluation of the models after applying the supervised fine-tuning settings described in Section 5.2.1.

### 5.3.1 LOSS COMPARISON

As shown in Figure 4, the training dynamics of FP8 and BF16 are virtually indistinguishable over the entire 160B-token continual pre-training trajectory. Both the validation loss (Figure 4a) and training loss (Figure 4b) exhibit nearly identical convergence patterns, with the FP8 curve (red) and BF16 curve (blue) remaining closely aligned throughout. The overlap between the two trajectories is so pronounced that they are almost visually indistinguishable, underscoring the numerical stability of FP8 training.

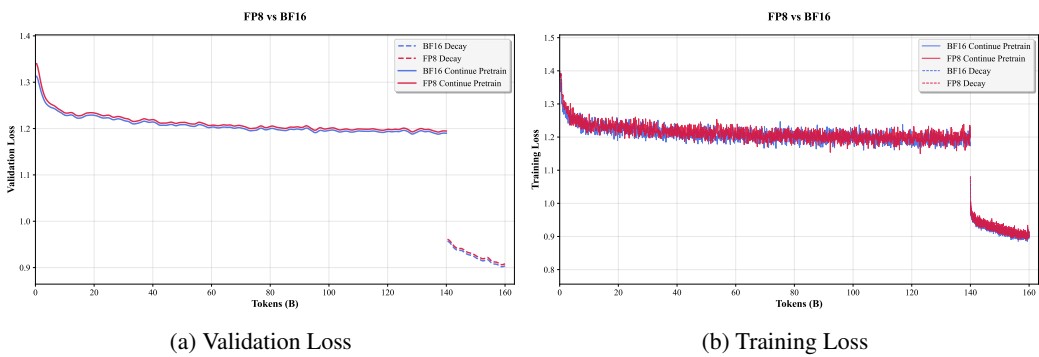

(a) Validation Loss          (b) Training Loss

Figure 4: The validation loss and training loss of Continue Pretraining Qwen2.5-1.5B-base comparing FP8 and BF16.

Table 5: Performance comparison of BF16 and FP8 training on reasoning benchmarks. All scores are reported as percentages (%).

|      | AIME 25 | AIME 24 | GPQA | LiveCodeBench v5 |
|------|---------|---------|------|------------------|
| BF16 | 17.91   | 17.50   | 31.94 | 16.41           |
| FP8  | 18.45   | 17.39   | 29.48 | 17.10           |

This finding provides strong empirical evidence that FP8 does not alter the underlying learning dynamics of the model. The equivalence between FP8 and BF16 in both optimization behavior and generalization performance demonstrates that our FP8 recipe preserves training fidelity, ensuring that efficiency gains are achieved without compromising convergence quality. Importantly, the consistency of the two curves across the full pre-training horizon suggests that FP8 training remains stable even under long-duration optimization at scale, reinforcing its viability as a drop-in replacement for BF16 in large-scale LLM training pipelines.

### 5.3.2 PERFORMANCE COMPARISON

Table 5 reports the performance of FP8 and BF16 training on mathematical and coding benchmarks. Notably, the FP8 results are obtained under a full FP8 training pipeline, covering both continual pretraining and subsequent 2-stage SFT. The results show that FP8 achieves accuracy on par with BF16 across all tasks. The small score variations indicate that reduced numerical precision introduces negligible degradation, confirming that FP8 training preserves the core reasoning capabilities of the model.

### 5.4 EFFICIENCY ANALYSIS

The performance evaluation presented in Table 6 highlights the substantial computational and memory efficiency gains achieved by our FP8 training compared to BF16 across different model configurations. The benchmarking encompasses both 1.5B and 7B parameter models under varying context lengths and parallelization strategies, providing a comprehensive view of FP8's effectiveness in practical large-scale training scenarios.

**Training Speed Optimization**: FP8 training reduces total training time by 7% to 22%, achieving a 22% speedup (0.78× ratio) for both model sizes at an 8k context length. This improvement is largely due to a reduction in backward pass computation time of up to 32%.

**Memory Efficiency**: FP8 consistently reduces peak memory consumption by 5% to 14%, reaching 11% (0.89× ratio) for the 1.5B model at a 32k context length. This memory saving facilitates the use of larger batch sizes and longer input sequences without additional hardware investment, thereby improving the effective utilization of existing GPU resources.

Table 6: Performance comparison of BF16 vs. FP8 quantization for Qwen2.5 model training. The table shows per forward/backward pass times, peak memory usage, and throughput across different model scales (1.5B, 7B) and context lengths (8k, 32k). FP8 achieves up to 22% faster training, 14% memory reduction, and 19% throughput improvement over BF16 baseline. TP: tensor parallelism, MBS: micro-batch size.

| | Forward | Backward | Total | Ratio | Peak Memory | Ratio | Throughput | Ratio |
|---|---|---|---|---|---|---|---|---|
| *Model Size = 1.5B* | | | | | | | | |
| Context Length = 32k, TP = 2, CP = 1, MBS = 1 | | | | | | | | |
| | Forward | Backward | Total | Ratio | Peak Memory | Ratio | Throughput | Ratio |
| BF16 | 841 ms | 2329 ms | 3170 ms | - | 57.8 GB | - | 345 TFlops | - |
| FP8 | 875 ms | 2075 ms | 2950 ms | 0.93× | 51.7 GB | 0.89× | 360 TFlops | 1.04× |
| Context Length = 8k, TP = 1, CP = 1, MBS = 2 | | | | | | | | |
| | Forward | Backward | Total | Ratio | Peak Memory | Ratio | Throughput | Ratio |
| BF16 | 463 ms | 1567 ms | 2030 ms | - | 68.1 GB | - | 340 TFlops | - |
| FP8 | 529 ms | 1061 ms | 1590 ms | 0.78× | 58.3 GB | 0.86× | 376 TFlops | 1.10× |
| *Model Size = 7B* | | | | | | | | |
| Context Length = 32k, TP = 4, CP = 1, MBS = 1 | | | | | | | | |
| | Forward | Backward | Total | Ratio | Peak Memory | Ratio | Throughput | Ratio |
| BF16 | 2790 ms | 6800 ms | 9590 ms | - | 78.1 GB | - | 409 TFlops | - |
| FP8 | 2660 ms | 5700 ms | 8360 ms | 0.87× | 67.4 GB | 0.86× | 461 TFlops | 1.14× |
| Context Length = 32k, TP = 2, CP = 1, MBS = 1 | | | | | | | | |
| | Forward | Backward | Total | Ratio | Peak Memory | Ratio | Throughput | Ratio |
| BF16 | 1760 ms | 5320 ms | 7080 ms | - | 53.2 GB | - | 453 TFlops | - |
| FP8 | 2300 ms | 3230 ms | 5530 ms | 0.78× | 50.8 GB | 0.95× | 537 TFlops | 1.19× |

**Computational Throughput Enhancement**: The method yields throughput improvements of 4% to 19%, with the 7B model at 8k context length showing a 19% gain (1.19× ratio), demonstrating effective of FP8 training.

Collectively, these results demonstrate that FP8 training achieves a highly favorable trade-off, delivering significant gains in training speed, memory efficiency, and throughput while maintaining model convergence and final performance. This establishes FP8 as a practical, scalable, and resource-efficient solution for large-scale language model training, providing tangible benefits for both research and production deployments.

## 6 CONCLUSION

In this work, we addressed the critical challenge of computational costs in LLM training by introducing and validating a comprehensive end-to-end FP8 training recipe. Our empirical results decisively demonstrate that FP8 training is both stable and effective. We have shown that learning dynamics, as reflected by training and validation loss curves, are nearly indistinguishable from the BF16 baseline. Furthermore, our FP8-trained models achieve performance on par with their BF16 counterparts across a range of challenging reasoning benchmarks, with performance differences typically within a 1–2 point margin attributable to evaluation noise.

These performance results are coupled with significant efficiency improvements, with up to 22% faster training, 14% lower memory usage, and 19% higher throughput. By providing this validated recipe and releasing key artifacts, including the high-performing InfiR2-7B-FP8 model which substantially outperforms its base model, we lower the barrier to entry for researchers and practitioners. Ultimately, our work establishes FP8 training not merely as a viable alternative, but as a compelling successor to BF16 for the next generation of LLMs, offering a sustainable path forward that balances cutting-edge performance with practical resource efficiency.

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

# A APPENDIX

## A.1 THE USAGE OF LLM

In this paper, a Large Language Model was utilized solely for the purpose of proofreading and improving the language and clarity of the manuscript. The core scientific contributions were performed exclusively by the authors.

