# OpenReview forum: "InfiR2: A Comprehensive FP8 Training Recipe for Reasoning-Enhanced Language Models"
_ICLR.cc/2026/Conference — ICLR 2026 Conference Withdrawn Submission_

### Official Review · Reviewer_ScwG · 2025-10-17

**Soundness:** 2
**Presentation:** 2
**Contribution:** 2
**Rating:** 4
**Confidence:** 4

**Summary:**

This paper introduces and validates an end-to-end FP8 training scheme, INFIR2, aimed at mitigating the high costs of LLM training. Through extensive experiments spanning continual pre-training and supervised fine-tuning, the authors demonstrate that this work achieves model performance on par with the BF16 baseline while yielding significant improvements in training speed, memory reduction, and throughput, thus offering a practical and efficient low-precision training solution for the community.

**Strengths:**

- The paper delivers an open-source, end-to-end FP8 training pipeline contributing to the community.
- The reported efficiency improvements are substantial.
- The study highlights the critical role of the UE8M0 scaling factor format in maintaining model performance during advanced training stages, offering insights into the stability of low-precision training.

**Weaknesses:**

- My main concerns lie in the generalizability of the proposed recipe across different model architectures. All experiments are strictly confined to the Qwen architecture, which undermines the core claim of a comprehensive recipe. Different architectures have variations in layer design and numerical distributions that may lead to different sensitivities to quantization strategies. For instance, the SwiGLU activations and RMSNorm used in Qwen2 might produce specific activation patterns for which the proposed hybrid quantization strategy is coincidentally well-suited. Without validation on a diverse set of popular architectures, it is difficult to ascertain whether this recipe is a truly universal solution or a specialized optimization for Qwen.
- The paper lacks an ablation study for its core hybrid-granularity quantization strategy, leaving it unclear why the specific combination of per-block for weights and per-token for activations is optimal.
- The largest model validated is only 7B parameters, which limits the extrapolation of the findings to much larger-scale models.
- The performance evaluation focuses primarily on reasoning benchmarks, lacking assessment on generative tasks that demand higher content fidelity.

**Questions:**

- What is the computational overhead of calculating the UE8M0 scaling factors compared to using FP32 scaling factors, and is this overhead consistent across different hardware?

---

### Official Review · Reviewer_3VeK · 2025-10-30

**Soundness:** 2
**Presentation:** 4
**Contribution:** 3
**Rating:** 6
**Confidence:** 2

**Summary:**

This paper presents INFIR2, a comprehensive, end-to-end FP8 training recipe designed to address the high computational cost of LLM training. It fills a key gap by providing an open-source methodology that integrates continual pre-training and supervised fine-tuning. The core method is a hybrid-granularity quantization strategy, using block-wise quantization for weights and higher-fidelity token-wise quantization for activations, while maintaining FP32 precision for optimizer states and master weights to ensure stability 4. The authors provide strong empirical validation, showing the recipe is "remarkably stable," with loss curves nearly identical to the BF16 baseline, and "essentially lossless," achieving on-par performance on reasoning benchmarks. This performance parity is coupled with substantial efficiency improvements, including up to 22% faster training, 14% lower peak memory, and 19% higher throughput.

**Strengths:**

1.	This paper proposes a solid recipe for unlocking end-to-end fp8 training, which combines different quantization recipes over model weights, activations and master weights e.t.c. It would mark a large step for community to employ fp8 training in daily development/
2.	Clear writing and good visualization

**Weaknesses:**

1. The paper's specific emphasis on "Reasoning-Enhanced" LLMs feels misplaced. The proposed FP8 recipe, a hybrid-granularity strategy of block-wise quantization for weights and token-wise for activations, appears to be a general, task-agnostic method for stable training. It is unclear how this *quantization method* is "specifically designed to preserve and enhance model reasoning" in a way that differs from how it would preserve any other capability (e.g., general knowledge or translation).

2. Following the first point, the choice of benchmarks (e.g., AIME24, GPQA) is used to prove the recipe is "lossless" on difficult problems. However, the paper equates task difficulty with high numerical sensitivity to quantization. This crucial link is not empirically established. The claims would be far more compelling with an ablation study. For instance, **does a simpler, "weaker" FP8 recipe (e.g., non-hybrid quantization) perform adequately on general benchmarks but fail on these reasoning tasks?** Without such a "sanity check," it's unclear if the complexity of the proposed hybrid-granularity recipe is truly necessary or if the "lossless" result could have been achieved with a simpler FP8 method.

3. The paper does not explicitly state the hardware used for its efficiency benchmarks. However, multiple references to the NVIDIA Hopper architecture and optimizations for "Deep GEMM operations" strongly imply the use of H100-series GPUs. This raises a significant concern about the generalizability of the results. The impressive efficiency gains (e.g., up to 22% faster training, 19% higher throughput) may be tightly coupled to Hopper's dedicated FP8 hardware support, which is not available on prior-generation Ampere GPUs. This limits the paper's stated goal of "democratizing large-scale model training” for the majority of the community that doesn’t have access to these advanced hardwares. Providing benchmark results on Ampere would substantially strengthen the paper's claims of practical, widespread utility.

4. The experiments, while thorough on 1.5B and 7B models, do not validate the recipe at a scale where FP8 stability is most critical. Intuitively, deeper and larger models (e.g., 30B+) are far more susceptible to **error accumulation** from low-precision arithmetic propagating through many more layers. While the stability shown at the 7B scale is promising (with loss curves nearly identical to BF16), this does not guarantee that the same numerical fidelity will hold at 30B, 70B, or higher scales. To convincingly demonstrate the recipe's robustness as a "drop-in replacement" for BF16 in large-scale training, validation on a significantly larger model is essential.

**Questions:**

See weaknesses.

---

### Official Review · Reviewer_X21F · 2025-10-30

**Soundness:** 3
**Presentation:** 2
**Contribution:** 2
**Rating:** 2
**Confidence:** 4

**Summary:**

The manuscript proposes an open-sourced, though without anonymous code provided, FP8 large language model post-training recipe. The authors claim that their FP8 training achieves no accuracy degradation and even performance gains for smaller language models.

**Strengths:**

1. A great engineering effort.
2. Some interesting findings regarding the use of lower-precision training on smaller models.

**Weaknesses:**

1. The main contribution of the manuscript appears to be the open-sourced training recipes, but no code is actually provided. I found a similar GitHub repository, InfiXAI/InfiR2, but its codebase seems immature and not ready for practical use. If the authors could provide more complete and usable artifacts for the community, the contribution would be much more significant.
2. The paper lacks a clear description of multi-GPU training. While lower-precision training is important, in commodity GPU servers, stable distributed training is of even higher priority. The manuscript does not mention this aspect, even in the setup section, making it unclear whether InfiR2 supports multi-GPU training. If not, limiting it to a single GPU would greatly reduce the benefits of using lower precision in large-scale training.
3. The experiments are restricted to post-training scenarios. It seems to suggest that lower-precision training is unstable or fragile when applied to pretraining from random initialization.
4. The hybrid granularity quantization is not newly proposed.

**Questions:**

1. Can InfiR2 be seamlessly integrated with different parallel training frameworks such as DeepSpeed?
2. Why not demonstrate system scalability by applying it to pretraining from scratch?
3. Could the authors provide more analysis on the benefits of FP8 training for smaller models? For example, does increased quantization error contribute to broader space exploration? Otherwise, the current contribution feels overly engineering-oriented.

---

### Official Review · Reviewer_wSkp · 2025-10-31

**Soundness:** 2
**Presentation:** 1
**Contribution:** 1
**Rating:** 2
**Confidence:** 4

**Summary:**

To reduce the computational cost of training large language models (LLMs), the authors propose an end-to-end FP8-based training recipe.
Their approach employs hybrid quantization granularity, applying per-block quantization to weights and per-token quantization to activations.
Experimental results demonstrate that the proposed FP8 training recipe achieves a 22% reduction in training time, a 14% decrease in peak memory usage, and a 19% increase in throughput, while maintaining performance comparable to BF16 training.

**Strengths:**

1. The paper provides a comprehensive and well-structured background on FP8 training in Section 3, which helps readers clearly understand the technical foundations of the proposed approach.
2. The authors conduct a thorough and systematic comparison between FP8 and BF16 training, offering detailed analyses of efficiency and performance differences.

**Weaknesses:**

1. The contribution of the paper could be articulated more clearly. It would be helpful for the authors to include a detailed comparison with related works such as COAT [1] and DeepSeek-V3 [2], highlighting differences in method design and results to better position their contribution.
2. The design rationale for the FP8 training recipe would benefit from deeper insights. Although the paper describes the hybrid quantization granularity, it does not explain the reasoning behind applying per-block quantization to weights and per-token quantization to activations. Including oracle experiments could help clarify these design choices.
3. The experimental section lacks an ablation study. A comparison of the proposed hybrid granularity strategy with alternative granularity settings would strengthen the empirical validation.
4. The presentation of experimental results is somewhat confusing. For example, Sections 5.1 and 5.3 appear to report nearly identical results, and Figure 3 is not referenced in the text. Improving the organization and discussion of results would enhance clarity.
5. There seems to be a color inconsistency in Figure 2—specifically, the “FP8 Per-token Quant” tensor does not appear within the FP8 block.

[1] Xi, Haocheng, et al. "Coat: Compressing optimizer states and activation for memory-efficient fp8 training." arXiv preprint arXiv:2410.19313 (2024).
[2] Liu, Aixin, et al. "Deepseek-v3 technical report." arXiv preprint arXiv:2412.19437 (2024).

**Questions:**

1. In Figure 4, why does the loss curve drop suddenly when the number of the training tokens exceeds 140B?

---

### Note · Authors · 2026-01-06

I have read and agree with the venue's withdrawal policy on behalf of myself and my co-authors.